# Barriers and facilitators for the use of telehealth by healthcare providers in India—A systematic review

Parth Sharma[1,2]*, Shirish Rao[1,3], Padmavathy Krishna Kumar[1,4], Aiswarya R. Nair[5], Disha Agrawal[1,2], Siddhesh Zadey[1,6,7,8], Gayathri Surendran[9], Rachna George Joseph[9], Girish Dayma[10], Liya Rafeekh[11], Shubhashis Saha[9], Sitanshi Sharma[12], S. S. Prakash[9], Venkatesan Sankarapandian[9], Preethi John[13], Vikram Patel[14]

1 Association for Socially Applicable Research (ASAR), Pune, Maharashtra, India, 2 Department of Community Medicine, Maulana Azad Medical College, Delhi, India, 3 Seth GS Medical College and KEM Hospital, Mumbai, Maharashtra, India, 4 Adichunchanagiri Institute of Medical Sciences, BG Nagara, Karnataka, India, 5 Travancore Medical College, Kollam, Kerala, India, 6 Dr D. Y. Patil Medical College, Hospital, and Research Centre Pune, Dr. D. Y. Patil Vidyapeeth, Pune, Maharashtra, India, 7 Global Emergency Medicine Innovation and Implementation (GEMINI) Research Center, Duke University School of Medicine, Durham, North Carolina, United States of America, 8 Department of Epidemiology, Columbia University Mailman School of Public Health, New York City, New York, United States of America, 9 Christian Medical College and Hospital, Vellore, Tamil Nadu, India, 10 KEM Hospital Research Centre, Pune, Maharashtra, India, 11 Indian Institute of Technology, Kharagpur, West Bengal, India, 12 Centre for Health Research and Development, Society for applied studies, Delhi, India, 13 Global Business School for Health, University College London, London, United Kingdom, 14 Harvard T.H. Chan School of Public Health, Boston, Maryland, United States of America

* parth.sharma25@gmail.com

## Abstract

It is widely assumed that telehealth tools like mHealth (mobile health), telemedicine, and tele-education can supplement the efficiency of Healthcare Providers (HCPs). We conducted a systematic review of evidence on the barriers and facilitators associated with the use of telehealth by HCPs in India. A systematic literature search following a pre-registered protocol (https://doi.org/10.17605/OSF.IO/KQ3U9 [PROTOCOL DOI]) was conducted on PubMed. The search strategy, inclusion, and exclusion criteria were based on the World Health Organization's action framework on Human Resources for Health (HRH) and Universal Health Coverage (UHC) in India with a specific focus on telehealth tools. Eligible articles published in English from 1st January 2001 to 17th February 2022 were included. One hundred and six studies were included in the review. Of these, 53 studies (50%) involved mHealth interventions, 25 (23.6%) involved telemedicine interventions whereas the remaining 28 (26.4%) involved the use of tele-education interventions by HCPs in India. In each category, most of the studies followed a quantitative study design and were mostly published in the last 5 years. The study sites were more commonly present in states in south India. The facilitators and barriers related to each type of intervention were analyzed under the following sub-headings- 1) Human resource related, 2) Application related 3) Technical, and 4) Others. The interventions were most commonly used for improving the management of mental health, non-communicable diseases, and maternal and child health. The use of telehealth has not been uniformly studied in India. The facilitators and barriers to telehealth

**Data Availability Statement:** Data file has been uploaded as supplementary file.

**Funding:** The author(s) received no specific funding for this work.

**Competing interests:** The authors have declared that no competing interests exist.

use need to be kept in mind while designing the intervention. Future studies should focus on looking at region-specific, intervention-specific, and health cadre-specific barriers and facilitators for the use of telehealth.

## Author summary

The use of telehealth has significantly increased in India over the past decade. Telehealth has the potential to address barriers of accessibility and affordability and can help in universalizing healthcare in India. In this review, we aimed to understand the barriers and facilitators to the use of telehealth by healthcare providers. We classified these into four categories—1) Human resource related, 2) Application related 3) Technical, and 4) Others. We also report the attitude of healthcare providers towards telehealth interventions and whether these interventions resulted in improvement of patient care and the performance of the healthcare provider. Understanding these barriers and facilitators is important as they will help in creating more contextually relevant telehealth policies in India that will help take India closer to the goal of universal health coverage.

## Introduction

Telehealth is defined as "the delivery and facilitation of health and health-related services including medical care, provider and patient education, health information services, and self-care via telecommunications and digital communication technologies" [1]. Even though used interchangeably, telehealth and telemedicine are not the same. Telehealth covers a wide range of services like telemedicine, mHealth (mobile health), and remote patient monitoring. Telemedicine refers to the delivery of diagnostic or treatment services to a patient using telecommunication technology remotely [1]. mHealth, on the other hand, refers to applications or programs used on smartphones or tablets [1] These interventions could be used to address the shortage of Health Care Providers (HCPs), for their education and training, or for supporting the functioning of the existing health workforce.

Access to healthcare of adequate quality is inequitable in India which disproportionately affects rural and low-resourced states [2], where a majority of the Indian population resides [3]. Access is worse for those belonging to vulnerable groups like the elderly, and people with disability [4]. A major driver for this inequitable access is the inequitable distribution of HCPs [5]. These barriers have resulted in the rapid privatization of healthcare in India [6], thus making healthcare a leading cause of out-of-pocket expenditure [7]. It is widely assumed that inequitable access to quality care could be addressed by telehealth interventions like mHealth and telemedicine and they could also help in cutting the out-of-pocket expenditure on healthcare [8–10]. Considering this, India has made efforts to incorporate telehealth into its health system.

To enhance the uptake of digital health interventions, the World Health Organization (WHO) published its Global Strategy on Digital Health for 2020–2025 [11]. In India, the National Health Policy 2017 recommended the use of Information and Communications Technologies to improve access to health services. In recent years, there has been a mushrooming of a range of telehealth interventions in India, for example, mSakhi and ASHA Kirana [12,13] in antenatal and postnatal maternal care through patient monitoring and behavior change communication; for the care of people with non-communicable diseases [14]; the

eSanjeevani telemedicine portal, a government-led initiative, to improve access, both in terms of affordability and accessibility, to care in remote areas [15], and to train HCPs. [16]

Considering the importance of telehealth in healthcare delivery in the Indian health system, this review was conducted with the primary objective of understanding the facilitators, and barriers associated with the use of telehealth tools, like telemedicine, tele-education, and mHealth, by HCPs in India. We also aimed to look at the role of telehealth in various aspects of the health system from service delivery, education, and training of HCPs, to its impact on their functioning and also their attitude toward the intervention.

## Methods

### Overview

This study is a systematic review conducted as one of the components of a larger evidence synthesis exercise undertaken by the Lancet Citizens' Commission on Reimagining India's Health System (www.citizenshealth.in). The protocol for evidence synthesis for the entire Human Resources for Health (HRH) workstream was registered on 16th June 2022 [17]. It complies with and the PRISMA guidelines [18] (S1 Checklist) and can be accessed here- https://doi.org/10.17605/OSF.IO/KQ3U9 [PROTOCOL DOI [19]. The objective of the larger evidence synthesis exercise mentioned above was to understand the HRH management strategies and practices for all cadres of human resources for health available in India. Since a large section of the papers were focusing on telehealth, this review was conducted to understand the barriers and facilitators to telehealth use by healthcare providers in India.

### Search strategy

The review was a part of the larger evidence synthesis work on HRH and their management for Universal Health Care (UHC) in India. The search was conducted for published literature between 1st January 2001 and 17th February 2022 in the PubMed database. The search strategy (S1 Panel) focused on the WHO action framework on HRH (S1 Table) that consists of six action fields (HR Management Systems, Leadership, Partnership, Finance, Education, and Policy) and four phases (Situational Analysis, Planning, Implementation, and Monitoring & Evaluation). This framework was used as it was developed to help governments develop and implement strategies to effectively manage the health workforce. Diverse categories of medical professional cadres (S2 Panel) along with universal health care in India were also added to the search strategy.

### Screening and selection

All the articles identified through the search strategy using the above-mentioned database were added to the DistillerSR software and duplicates were removed. A multi-level screening of articles using DistillerSR software was carried out by the team as described in the PRISMA diagram (**Fig 1**). Inclusion criteria included studies conducted in India and reported in English that focused on the use of telehealth by healthcare providers. Studies only evaluating clinical outcomes but not related to HRH cadre or management strategies or practices and study protocols, editorials, viewpoints, commentaries, letters, and correspondences were excluded.

The articles were divided into a team of two reviewers. At Level 1, the articles were screened based on the title and abstract. The articles included by any one reviewer at Level 1 screening were moved to Level 2. The full text of all the articles in Level 2 was reviewed independently by two reviewers. After the full-text screening, articles were finally excluded or included only if

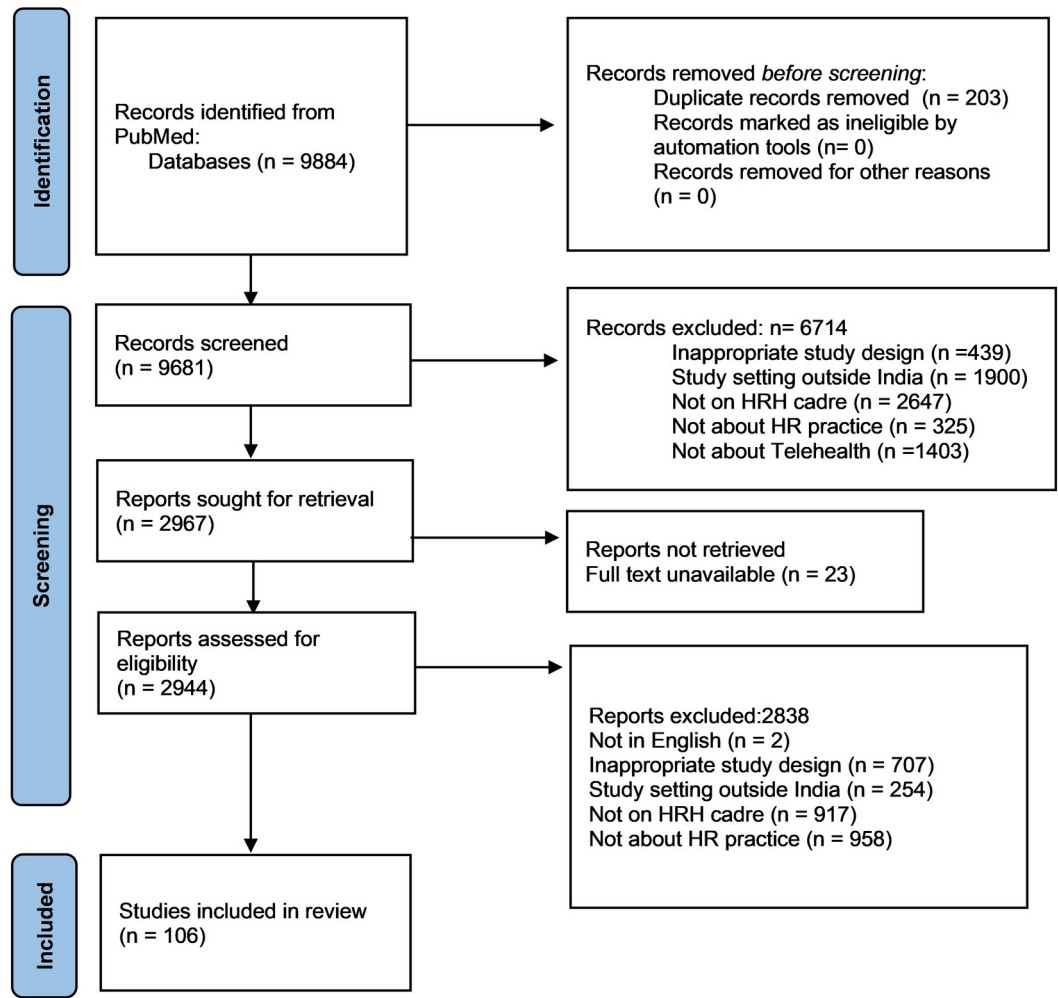

**Fig 1. PRISMA flowchart showing the selection and inclusion of the studies in the review.**

both reviewers were in agreement. Conflicts about the eligibility criteria were resolved either through consensus between the two reviewers or by consulting one more reviewer.

## Data extraction, analysis, and risk of bias assessment

At Level 3, data charting for all the included full-text articles was done. Charting done by one author was verified by the other author. Text from the manuscripts was extracted under the following headings: Authors, Year of Publication, Study Design, Study Setting, Study Location, Human Resource (HR) cadre, HR Practice, Sample Size, Primary Objectives, Primary Outcomes, Impact, Challenges and Barriers, and Study Limitations. Articles were then classified based on the type of telehealth intervention into mHealth, telemedicine, and tele-education. Article characteristics (year of publication, study design, study setting, and location) and HR characteristics (HR cadre, and practice) were summarized as frequency and percentages. Manual thematic analysis was done to identify relevant sub-themes of facilitators and barriers under the following broader themes—human resources-related, application-related, and technical. Sub-themes not fitting under these three broad themes were classified as others. Similarly, a manual thematic analysis was done to identify the impact of intervention on the

healthcare worker (S1 Data). The frequencies of themes under each section were reported. The role of the intervention and limitations of the study were extracted as mentioned by the authors and their frequencies were reported. The quality appraisal of the studies was assessed using Joanna Briggs Institute's (JBI) Critical Appraisal Tools. Two reviewers independently evaluated each included study using the JBI Critical Appraisal Tool that was appropriate for the particular study design. Quality scores were based on the following study domains, including eligibility criteria, participant characteristics, and data analysis among others. Each checklist domain was evaluated independently by two co-authors. Conflicts in evaluation were settled through discussion among the reviewers. When agreement could not be obtained, the final judgment was determined after consulting a third reviewer. Rating scales of 'Yes,' 'No,' 'Unclear,' and 'Not Applicable' were used to assess each question of the tool. A "Yes" response was given one point, a "No" response was given zero points, and an "Unclear" response was given half a point. The percentage of the final score was calculated to classify the studies into low, moderate, and high risk of bias. A score of less than 50% was rated as high risk of bias, 50 to 69% as moderate, and more than 70% as low risk of bias. The mixed-method studies were appraised using the Mixed Methods Appraisal Tool (MMAT) Version 2018.

## Results

One hundred and six studies were included in the review. Of these, 53 studies (50%) involved mHealth interventions [13, 20–71], 25 (23.6%) involved telemedicine interventions [72–96] whereas the remaining 28 (26.4%) involved the use of tele-education interventions by HCPs in India [97–124]. On risk of bias assessment, 94.5% of the cross-sectional studies, 53.3% of cohort studies, 44.4% of randomized controlled trials, 100% of qualitative studies, 80% of review studies, and 88.9% of quasi-experimental studies had low risk. The remaining studies had a moderate risk of bias. No included study had a high risk of bias. A detailed assessment of the risk of bias for each study has been presented in S2 Table.

### mHealth

Of the total 53 studies, nearly half the studies (45%) were quantitative [21,26–28,31,33–36,38,39,42,43,45,50,51,53,55,58,59,63,64,66,70], 14 (26%) were qualitative [13,20,22,29,32,37,44,46–48,60,65,67,69], 12 (23%) were mixed-methods [23–25,30,40,41,52,54,56,57,62,68] and 3 (6%) were review studies [49,61,71]. No study on the use of mHealth by an HCP was published before 2013, and 64% [13,20–51,55] of the studies were published after 2018, with the maximum (n = 12, 23%) [13,22,24–31,33,34] number of studies being published in 2021. The studies were conducted in tertiary care or teaching hospital settings (n = 28) [22,23,26–29,31–38,46,47,49,55–59,61,63,65,68,69,71], community health centers (n = 6) [20,21,42,44,60,66], primary health centers (n = 16) [13,30,39,40,43,48,50–54,62,64,66,67,70], and other settings (n = 4) [24,25,41,45]. The use of mHealth was most studied in Karnataka (n = 7) [13,28,41,52,66,67,71], followed by Gujarat [21,38,44,58,65], Maharashtra [26,28,32,49,66], and Tamil Nadu [26,36,37,51,53] (5 studies each) (**Fig 2**). Findings from all included studies have been summarized in the **S3 Table**.

mHealth interventions were most commonly used by doctors (n = 38) [21–23,25–27,30,31,35–38,40–45,47–50,52–54,56–62,64,65,68–71], followed by community healthcare workers (n = 18) [22,28–30,32–34,36,39,41,44,46,49,51,58,63,66,67], nurses (n = 8) [13,20,22,41,42,55,56,68], allied health professionals (n = 4) [43,54,68,70], auxiliary midwife nurses (n = 3) [22,36,41], and others (n = 8) [24,29,36,46,54,62,64,68].

**1. Facilitators and barriers to the use of mHealth.** Prior training to use the mHealth intervention (n = 19) [13,20,23,25,26,32,37,41,42,44,46,49,50,52,55,57,60,62,63], interactive

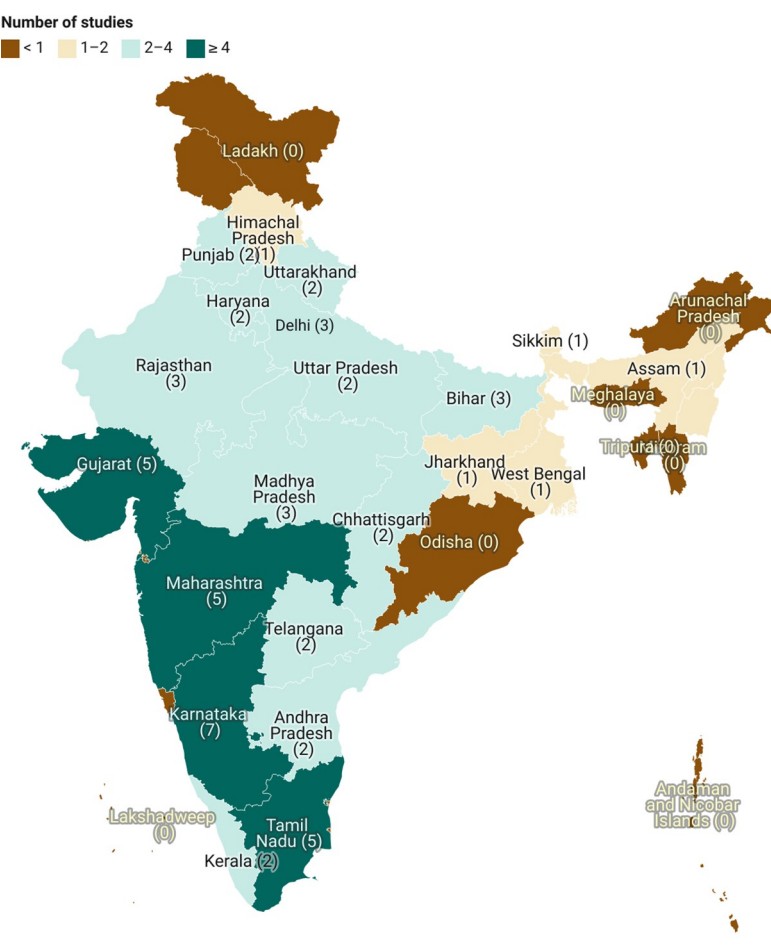

Map data: © OSM · Created with Datawrapper

**Fig 2. The number of study sites per state in India for mHealth (The map was created using app.datawrapper.de, an open source online software.** The copyright of the map belongs to the author).

intervention with the use of videos and images (n = 14) [21,23,34,39–41,44–46,49,60,67–69], and availability of the device to use the intervention (n = 6) [22,23,48,49,58,61] were the most common human resource-related, application-related, and technical facilitators respectively. Formative research before designing the intervention (n = 4) [29,32,44,48] and government support for the intervention (n = 2) [29,30] were other facilitators that were identified. Other facilitators are mentioned in **Fig 3**.

Low digital literacy (n = 10) [13,22,26,32,38,41,44,46,65,69], malfunctioning of the software (n = 13) [13,20–22,24,25,29,31,37,43,66,67,69], and poor network connectivity (n = 14) [20–22,24,32,34,38,41,44,47,53,55,67,69] were the most common human resource-related, application-related, and technical barriers respectively. Stigma related to technology (n = 4) [13,38,68,69], worsening of disease-related stigma due to the use of technology (n = 3) [41,55,62], lack of formative research (n = 1) [69], and lack of human touch due to the use of mHealth (n = 1) [34] were other barriers that were identified. Other barriers are mentioned in **Fig 4**.

**2. Role of mHealth.** The mHealth interventions were most commonly used for improving maternal and child healthcare (n = 24) [13,20,22–25,27,33,34,36,39–43,53,54,59,60,63,64,66–68], followed by non-communicable diseases like diabetes, hypertension, and cancer (n = 12)

| Human Resource Related | Application-related | Technical | Others |
| --- | --- | --- | --- |
| • Prior training to use mhealth (13,20,23,25,26,32,37,41,42,44,46,49,50,52,55,57,60,62,63)<br>• Additional technical support (25,32,39,40,46)<br>• HCP with better relationship with the community (30,51)<br>• Incentives for mhealth use (64,69)<br>• Opportunity to HCW to get involved in patient care (21) | • Ease of using and adaptability of the intervention (20,21,23,25,29,37,38,41,42,48,52,67)<br>• Interactive app with videos and images (21,23,34,39–41,44–46,49,60,67–69)<br>• Use of local language (23,25,41,44,50,54,55,57,60)<br>• Cost effectiveness of the intervention (31,37,38,55,56)<br>• Offline content in the app (32,35,45)<br>• An intervention that does not increase burden (24,39,40) | • Device availability (22,23,48,49,58,61)<br>• Satellite connectivity (32,35,45)<br>• Device with longer battery life and better functionality (23,42) | • Formative research to support fit with the context and population (29,32,44,48)<br>• Government support for intervention (29,30) |

**Fig 3. Facilitators of use of mHealth.**

[26,28,30,31,46,47,50–52,55,57,65] and mental health (n = 6) [21,32,44,49,61,62]. Based on the WHO action framework on HRH, 29 (55%) studies focused on HR management and aimed at improving the efficiency of available human resources [13,22–24,28–30,33,35–40,42,43,46,51,54,55,57,58,61,64–67,69,71]. Twenty-three (43%) studies involved mHealth interventions that aimed at the education and training of HCPs [20,21,25–27,31,32,34,41,44,45,47–50,52,53,59,60,62,63,68,70] Only one study looked at the financial aspect of the intervention's use by the HCPs [56].

**3. Impact of mHealth Interventions and Attitude of HCPs towards Them.** The use of mHealth impacted the practice of HCPs in various ways. Improvement in patient outcome was reported in 22 studies [20,24,28,33,36,39–43,46,47,49,51,53,54,60–63,69,71], improvement in knowledge of HCP in 18 studies [13,20,21,23,27,29,36,41,44,45,53,55,59,62,65,68–70], and improvement in work performance of HCP in 24 studies [13,20,22–25,29,33,38,40–

| Human Resource-related | Application-related | Technical | Others |
|---|---|---|---|
| • Low digital literacy (13,22,26,32,38, 41,44,46,65,69)<br>• Shortage of HCP (20,22,46,66,69)<br>• Lack of awareness regarding app functions (20,21,31,32,68)<br>• Lack of training or poor quality of training (21,22,26,66,69)<br>• Lack of motivation or short-lasting interest (26,67)<br>• Data safety and legal concerns (37,38,58)<br>• Fear of internet addiction (65,68)<br>• Difficulty in communication while using mHealth (38)<br>• Lack of technical support (29)<br>• Time constraints during high workload (20) | • Malfunctioning of the application (13,20–22,24,25,29,31,37,43,66,67,69)<br>• Difficult to understand the language used in the application (20,21,25,34)<br>• Difficulty in using the application (21,34,67)<br>• Lack of interoperability between different mHealth tools. (29,69) | • Poor network (20–22, 24, 32, 34,38,41,44,47,53,55,67,69)<br>• Malfunctioning of device (20,21,23–25, 42–44,67,69)<br>• Lack of access to device (22,25,26,29,41, 68)<br>• Poor infrastructure - lack of electricity, battery backup (22,69) | • Stigma related to the technology (13,38,68,69)<br>• Stigma related to the disease worsening due to technology use (41,55,62)<br>• Lack of human touch (34)<br>• Lack of formative research (69) |

**Fig 4. Barriers to the use of mHealth.**

43,45,46,48,52–54,59,65,67,69,71]. Studies also reported an improvement in confidence (n = 7) [13,20,23,42,46,52,68] and communication (n = 7) [13,40,41,43,50,54,58] while using mHealth interventions. The other impacts are mentioned in **Table 1**.

**Table 1. Impact of telehealth interventions.**

| Variable | mHealth | Telemedicine | Tele-education |
|---|---|---|---|
| | Number of studies (n = 53) | Number of studies (n = 25) | Number of studies (n = 28) |
| Improvement in work performance | 24 [13,20,22–25,29,33,38,40–43,45,46,48,52–54,59,65,67,69,71] | 3 [76,85,92] | 5 [97,101,105,114,124] |
| Improvement in patient outcome | 22 [20,24,28,33,36,39–43,46,47,49,51,53,54,60–63,69,71] | 16 [74–78,81,82,85–87,89, 91–94,96] | 2 [102,110] |
| Improvement in knowledge of HCP | 18 [13,20,21,23,27,29,36,41, 44,45,53,55,59,62,65,68–70] | 3 [76,85,89] | 17 [97,100,101,104,105, 107,108,110,111,114–118,120,122,124] |
| Increases social status/ recognition of work/care seeking/trust/reliability of HCP | 11 [13,22,23,25,36,39–41,43,52,69] | 2 [76,82] | - |
| Promotes better communication and relationship between HCP-HCP/HCP-patient | 7 [13,40,41,43,50,54,58] | 2 [76,95] | 1 [119] |
| Increase in confidence | 7 [13,20,23,42,46,52,68] | 1 [76] | 5 [104,107,110,111,115] |
| Flexibility to learn offered by the intervention | 7 [21,23,29,31,44,65,70] | 1 [83] | 4 [116,118,122,123] |
| Saves time | 6 [20,41,44,48,52,71] | 3 [87,92,94] | - |
| No diagnostic difference as compared to conventional techniques | 3 [47,57,61] | 3 [79,88,92] | 1 [124] |
| Decrease in workload/stress | 3 [20,39,46] | 2 [76,94] | - |
| Decreases travel | 2 [44,56] | 4 [78,91,92,94] | 1 [101] |
| Increased motivation of HCP due to the intervention | 2 [52,69] | 1 [77] | 1 [122] |

Out of the 53 studies, 26 studies reported positive attitudes of HCPs toward mHealth interventions [13,22,25,26,29,31,32,34–38,40–46,49,52,53,55,58,68,69] whereas 1 study reported a negative attitude [65] and the remaining did not mention the attitude of the HCP toward the intervention. Nine studies reported that the HCP was satisfied with the intervention [24,29,31,32,43,54,55,61,68], and in two studies, the HCP mentioned that they would recommend the intervention to others [23,31].

**4. Limitations of studies assessing the use of mHealth.** Studies evaluating the use of mHealth interventions commonly cited inadequate sample size (n = 10) [21,29,32,38–40,44,62,64,69], poor sampling techniques (n = 9) [26,36,38–40,44,62,64,69], and incomplete data (n = 7) [33,39,40,42,47,50,63] as limitations. Desirability bias, as mentioned in 8 studies [20,21,23,25,32,36,39,64], could have resulted in a more positive outcome of the interventions being studied. Other limitations of studies are mentioned in **Table 2**.

**Table 2. Limitations of studies included in the review.**

| Variable | mHealth | Telemedicine | Tele-education |
|---|---|---|---|
| | Number of studies (n = 53) | Number of studies (n = 25) | Number of studies (n = 28) |
| Inadequate sample size | 10 [21,29,32,38–40,44,62,64,69] | 4 [74,81,85,86] | 4 [107,110,111,114] |
| Poor sampling technique | 9 [26,36,38–40,44,62,64,69] | 2 [72,73] | 3 [114,121,123] |
| Incomplete data/other data-related constraints | 7 [33,39,40,42,47,50,63] | 2 [74,87] | 2 [106,108] |
| Poor study design | 6 [28,49,51,56,63,65] | 4 [80–82,96] | 2 [109,123] |
| Assessed perception only and not hard outcomes | 6 [20,26,31,36,42,62] | - | - |
| Short-term effect only assessed | 2 [31,68] | - | 3 [106,107,116] |
| Short duration of the study | 3 [30,36,43] | - | 1 [115] |
| Resource constraints | 2 [23,35] | - | - |
| Desirability bias<br>Recall bias<br>Hawthorne bias | 8 [20,21,23,25,32,36,39,64]<br>2 [43,56]<br>1 [33] | 1 [83]<br>1 [73]<br>- | - |
| Inappropriate study setting | 1 [44] | - | - |

## Telemedicine

Twenty-one studies (84%) were quantitative [72–77,79–82,84–89,91,93–96], 2 (8%) were qualitative [78,83], 1 (4%) followed mixed methodology [90] and 1 (4%) was a review study [92]. No study on the use of telemedicine by an HCP was published before 2011. A majority (64%) [74–78,82,84–87,89,91–93,95,96] of the studies were published after 2017 with the maximum (n = 5, 20%) [75,77,84,86,91] number of studies being published in 2020. Nearly all the studies were conducted in tertiary care settings or teaching hospital settings (92%) [72,73,75–88,90–96], and only 1 study each was conducted in primary health centers [81], community health centers [73], HIV clinics [74], and non-governmental organization clinics [89]. The use of telemedicine interventions was most studied in Karnataka (n = 5) [72,75,77,94,96] followed by Andhra Pradesh (n = 3) [73,76,93] and Bihar (n = 2) [86,87] (**Fig 5**). Findings from all included studies have been summarized in the **S4 Table**.

Telehealth was most commonly used by doctors (n = 19) [72–77,79,80,82,83,85–88,90,91,93,94,96] and focussed more on nurses (n = 5) [82,84,86,89,96] than community healthcare workers (n = 3) [72,77,95], allied health professionals (n = 2) [72,89], and auxiliary midwife nurses (n = 2) [72,81].

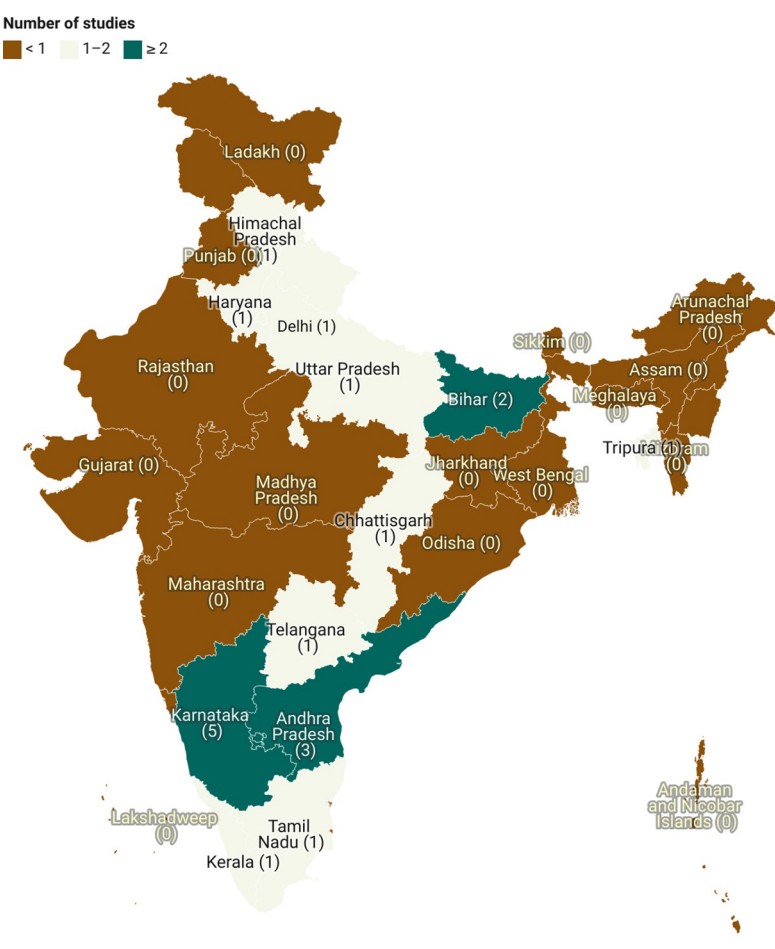

**Fig 5. The number of study sites per state in India for telemedicine (The map was created using app.datawrapper. de, an open source online software.** The copyright of the map belongs to the author).

| Human Resource Related | Application-related | Technical |
|---|---|---|
| • Prior training to use telemedicine (75,93)<br>• Additional technical support (76) | • Ease of using and adaptability of the intervention (92)<br>• Use of local language (89)<br>• Cost effectiveness of intervention (74,75,78,85,87,93,94) | • Satellite connectivity (88) |

**Fig 6. Facilitators of use of telemedicine.**

**1. Facilitators and barriers to the use of telemedicine.** Prior training to use the telemedicine intervention (n = 2) [75,93], use of local language (n = 1) [89], and additional technical support (n = 1) [76] were identified to be the human resource-related facilitators. The availability of satellite connectivity (n = 1) [88] was a technical facilitator that improved the uptake of telemedicine. Cost-effectiveness (n = 7) [74,75,78,85,87,93,94], and ease of use of the intervention (n = 1) [92] were the application-related facilitators (**Fig 6**).

Poor network connectivity (n = 8) [73,76,81,84,87,92,94,96], difficulty in understanding English, the language used in the application (n = 5) [84,86,87,92,93], and difficulty in communicating while using telemedicine (n = 6) [76,83,86,87,93,95] were the most common technical, application-related, and human resource-related barriers respectively. Lack of human touch (n = 5) [77,80,83,91,95] and stigma related to technology (n = 1) [94] also acted as barriers to the uptake of telemedicine. Other barriers are mentioned in **Fig 7**.

**2. Role of telemedicine.** Telemedicine was most commonly used for providing treatment for conditions related to maternal and child health (n = 5) [76,80,82,84,86], non-communicable diseases (n = 3) [88,91,94] like diabetes, hypertension, and cancer, and mental health (n = 3) [77,78,95]. While most studies focused on improving the efficiency and performance of the HCP (n = 23) [73–89,91–96], 1 study focused on the knowledge and awareness regarding telemedicine in the HCPs [72] and 1 study addressed the policy and financial aspects of telemedicine [90]

**3. Impact of telemedicine interventions and attitude of HCP towards them.** Improvement in patient outcome (n = 16) [74–78,81,82,85–87,89,91–94,96], improvement in knowledge of HCP (n = 3) [76,85,89], and improvement in work performance (n = 3) [76,85,92] were associated with the use of telemedicine. It also helped in reducing travel (n = 4) [78,91,92,94] and when used for remote diagnosis, telemedicine showed no significant diagnostic difference when compared with conventional diagnostic modalities (n = 3) [79,88,92]. The other impacts are mentioned in **Table 1**.

Out of the 25 studies, 13 studies reported positive attitudes of HCPs toward telemedicine interventions [76–78,83,85–87,91–96] whereas 1 study reported a negative attitude [90] and the remaining did not mention the attitude of the HCP toward the intervention. Twelve studies reported that the HCP was satisfied with the intervention [76,77,82–84,86,87,91–94,96],

| Human Resource-related | Application-related | Technical | Others |
|---|---|---|---|
| • Low digital literacy (72,76,94,95)<br>• Difficulty in communication while using telemedicine (76,83,86,87,93,95)<br>• Shortage of HCP (77,78,92,93)<br>• Data safety and legal concerns (72,85,90,94)<br>• Lack of motivation or short-lasting interest (74)<br>• Lack of training or poor quality of training (72)<br>• Time constraints during high workload (73) | • Malfunctioning of the application (81,83,93)<br>• Difficult to understand the language used in the application (84,86,87,92,93)<br>• Difficulty in using the application (96) | • Poor network (73,76,81,84,87, 92,94,96)<br>• Malfunctioning of device (76,81,92)<br>• Lack of access to device (84,86,95,96)<br>• Poor infrastructure - lack of electricity, battery backup (92–94) | • Lack of human touch (77,80,83,91,95)<br>• Stigma related to the technology (94)<br>• Unstable patients could not be managed (77,86,87)<br>• Difficult to track progress of patients (75,82,85,91) |

**Fig 7. Barriers to the use of telemedicine.**

and in 2 studies, HCPs mentioned that they would recommend the intervention to others [86,87].

**4. Limitations of studies assessing the use of telemedicine.** Inadequate sample size (n = 4) [74, 81, 85, 86], poor study design (n = 4) [80–82, 96], and poor sampling techniques (n = 2) [72, 73] were the most commonly cited limitations in the studies included. Other limitations of studies are mentioned in **Table 2**.

## Tele-education

Twenty-four studies (85.7%) were quantitative [97–102,105–109,111,112,115–124], 2 (7.1%) were qualitative [104,113], 1 (3.6%) followed mixed methodology [114] and 1 (3.6%) was a review study [103]. No study on the use of tele-education was published before 2009. A majority (72%) [97,98,100–102,104–109,111–113,116–118,121–123] of the studies were published after 2017 with the maximum (n = 10, 36%) [97,98,101,106,108,109,117,121–123] number of studies being published in 2021. Nearly all the studies were conducted in tertiary care settings or teaching institutes (86%) [98–100,102–107,109–117,119–124], and only 3 studies were

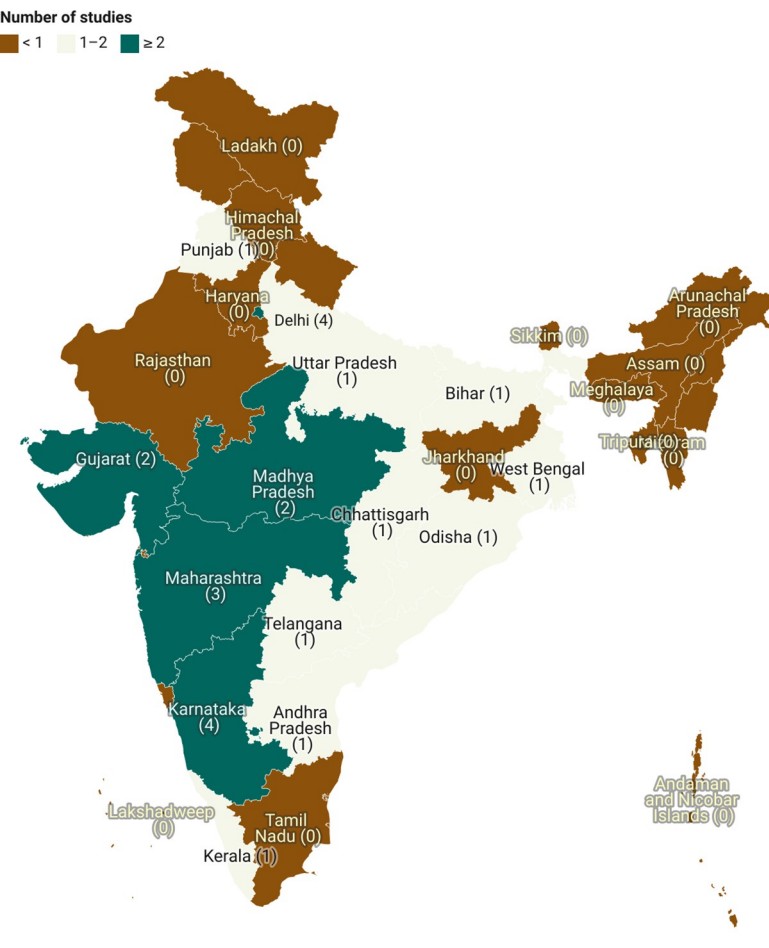

**Fig 8. The number of study sites per state in India for tele-education (The map was created using app. datawrapper.de, an open source online software.** The copyright of the map belongs to the author).

conducted in primary health centers [101,108,118] and 2 in community health centers [101,108]. The use of tele-education was most studied in Karnataka (n = 4) [100,104,111,112] and Delhi (n = 4) [102,111,117,120] (**Fig 8**). Findings from all included studies have been summarized in the **S5 Table**.

Tele-education services were most commonly meant for doctors (n = 16) [97–103,106,109–111,117–119,121,124] followed by nurses (n = 9) [98,105,106,109,111,117,121,122,124], community healthcare workers (n = 8) [103–105,107,114–116,123], allied health professionals (n = 5) [98,109,114,121,124], and auxiliary midwife nurses (n = 4) [108,111,112,121].

**1. Facilitators and barriers to the use of tele-education.** Similar to telemedicine, prior training to use the tele-education intervention (n = 2) [102,118] and ease of using the intervention (n = 2) [112,122] were the most common human resource-related, and application-related facilitators respectively. Availability of a device (n = 2) [110,118] was identified to be a technical facilitator. Formative research before designing the intervention (n = 1) [104] also helped in increasing its uptake as the formative research helped in addressing the needs of the participants (**Fig 9**).

Similar to telemedicine, low digital literacy (n = 2) [104,115], and poor network connectivity (n = 11) [98–100,103,104,109,111,113,117,120,122] were the most common human

| Human Resource Related | Application-related | Technical | Others |
|---|---|---|---|
| • Prior training to use tele-education (102,118) | • Ease of using and adaptability of the intervention (112,122)<br>• Interactive app with videos and images (117)<br>• Use of local language (113,118)<br>• Cost effectiveness of intervention (101) | • Device availability (110,118) | • Formative research to support fit with the context and population (104) |

**Fig 9. Facilitators of use of tele-education.**

resource-related, and technical barriers respectively. Difficulty in understanding English (n = 2) [111,114], the language commonly used for the applications, and malfunctioning of the software (n = 2) [111, 113] were application-related barriers. Other barriers are mentioned in **Fig 10**.

**2. Role of tele-education interventions.** Tele-education services were most commonly used for educating about mental health disorders (n = 9) [97,98,101,102,104,107,109,111,121] followed by non-communicable diseases like diabetes, hypertension, and cancer (n = 4) [100,113,116,118] and maternal and child healthcare (n = 4) [104,114,115,120]. Two studies each focused on educating about oral health problems [105,106] and HIV [110,114] and one study addressed teleteaching for orthopedics [112], critical care [108], COVID-19 [117], palliative care [122] and cardiology [124] Four studies did not mention what tele-education was used for [99,103,119,123].

**3. Impact of tele-education Interventions and Attitude of HCP towards Them.** Tele-education resulted in an improvement in knowledge of HCP (n = 17) [97,100,101,104,105,107,108,110,111,114–118,120,122,124] and an improvement in the work performance of HCP (n = 5) [97,101,105,114,124]. Its use also resulted in improvement in the confidence (n = 5) [104,107,110,111,115] and communication (n = 1) [119] of HCPs. The other impacts are mentioned in **Table 1**.

Out of the 28 studies, 15 studies reported positive attitudes of HCPs toward tele-education interventions [97,100,104,105,108–112,114–116,122–124] whereas 1 study reported a negative attitude [99] and the remaining did not mention the attitude of the HCP toward the intervention. Nine studies reported that the HCP was satisfied with the intervention

| Human Resource-related | Application-related | Technical | Others |
|---|---|---|---|
| • Low digital literacy (104,115)<br>• Difficulty in communication while using tele-education (123)<br>• Shortage of HCP (107,117)<br>• Lack of motivation or short-lasting interest (105,112,113)<br>• Lack of training or poor quality of training (99)<br>• Lack of technical support (117) | • Malfunctioning of the application (111,113)<br>• Difficult to understand the language used in the application (111,114) | • Poor Network (98–100, 103, 104,109,111,113,117,120,122)<br>• Malfunctioning of device (114,119)<br>• Lack of access to device (99)<br>• Poor infrastructure - lack of electricity, battery backup (117) | • Stigma related to the technology (99,105)<br>• Stigma related to the disease worsening due to technology use (114)<br>• Difficult to keep track of participants (108,113,120) |

**Fig 10. Barriers to the use of tele-education.**

[97,107,110,111,114,115,119,120,124] and in 3 studies, HCPs mentioned that they would recommend the intervention to others [110,115,122].

**4. Limitations of studies assessing the use of tele-education.** The limitations of the studies included were similar to the limitations cited by studies that assessed telemedicine with inadequate sample size (n = 4) [107,110,111,114], poor study design (n = 2) [109,123], and poor sampling techniques (n = 3) [114,121,123] being the most commonly cited limitations. Other limitations of studies are mentioned in **Table 2.**

## Discussion

This systematic review looks at the facilitators and barriers to the application of telehealth for various HCP Practices in the Indian health system. Even though a wide variety of interventions in the form of mHealth, telemedicine, and tele-education have been explored, only 8 states/union territories were the sites for most of the interventions. The use of telehealth by doctors, nurses, and community health workers was commonly addressed and literature on the use of the same by allied health professionals and non-medical healthcare workers was limited. Telehealth was most commonly used for HRH management aiming to improve the efficiency of available human resources. Maternal and child health, non-communicable diseases like diabetes, hypertension, obstructive airway disease, and cancer, and mental health were common areas of focus for the use of telehealth. Few studies looked at the use of telehealth for the provision of acute medical care, follow-up of patients after discharge, provision, and monitoring of home-based palliative care, and improvement in treatment compliance of patients with HIV

and tuberculosis. Studies conducted globally, have also assessed the utility of telemedicine, mHealth, and tele-education for similar diseases and conditions as done by the studies in India [125–127] A systematic review by Braun R. et al. conducted in 2013, which included 25 articles found that community health workers used mHealth to advance a broad range of health aims throughout the globe, particularly maternal and child health, HIV/AIDS, and sexual and reproductive health [125] Another scoping review by Lee Y et al., emphasized the utility of mHealth and telemedicine for tuberculosis control. The review included 145 studies published between 2016–19 of which almost 74% of the studies focused on its use by HCPs and about 50% were from the United States (21%), China (14%), and India (12%) [127].

This review brings to light multiple facilitators and barriers to telehealth adoption and use. The findings could help in the modification of national policies and guidelines which currently are not very robust [128]. An understanding of the facilitators and barriers aids in understanding the need for policy modifications at multiple dimensions, especially focusing on HCPs, Infrastructure, and Technology. Moreover, the facilitators and barriers identified for mHealth, telemedicine, and tele-education are similar to each other and those reported previously in the context of telehealth in Lower Middle-Income Countries (LMICs). Technical and infrastructural barriers in the form of internet access, device access, connectivity issues, poor battery life, and unstable electricity supply have been reported to contribute to major barriers in implementing telehealth services in LMICs by studies conducted in Sub-Saharan African and Middle Eastern countries [129–131]. This is especially important in the context of India, where over 70% of the population resides in rural areas, which are highly vulnerable to the aforementioned barriers [132] In terms of barriers, previous reviews conducted for Sub-Saharan African and Middle Eastern countries have identified HCP shortage, insufficient training and skills, additional workload, lack of motivation, lack of technical support, lack of integration with other government systems, and data safety and legal concerns [129–131,133,134] Additionally, our study provides deeper insights into barriers faced by the provider like fear of internet addiction, language barriers, and malfunction of applications. Barriers concerning the lack of human touch and stigma related to subpar patient care have also been previously raised by a systematic review conducted by Kruse C. S. et al [135].

Historically, previous reviews have reported financial barriers in the form of sponsorships and funding, capital expenses for technology start-up and maintenance, and budget constraints [129,131]. However, in the Indian context, government support and funding for telehealth interventions were an important facilitator for their implementations as reported by 2 studies included in our review [29,30] Previous studies have also shown that a strong commitment from the governments towards supporting and financing telehealth has been one the major facilitators [134]. However, funding towards health overall is still largely limited in India as only 2.1% of the gross domestic product is invested in the public healthcare sectors. The underfunding has thus resulted in digital health largely being neglected as more pressing issues like immunization and the provision of maternal and child care take priority [136]. HCP & application-related facilitators in the form of prior training, technical support, use of local language, and better user interface, as reported by Ag Ahmed MA et al. [134], were also reported in over one-third of the studies from our review. Additionally, providing incentives for telehealth use, use of offline material, balanced overload, and the relationship of community health workers with the community were also found to be other important facilitators in our review. Formative research to support fit with the context and population was seen as an important facilitator for telehealth in India; this emphasises the need for regional research as well as customizing the intervention as per the setting. Fifteen studies also emphasized the cost-effectiveness of telehealth interventions, which serve as a vital facilitator in resource-constrained settings like India.

Our review reported a strong impact of telehealth on patient care in terms of better patient outcomes, treatment compliance, and disease knowledge. It reduced travel constraints and improved accessibility for both patients and healthcare providers which has also been shown to improve the previously mentioned outcomes [137,138]. Specifically for healthcare workers, a greater number of studies showed that the use of telehealth improved their performance, confidence, and patient communication. Globally as well, multiple studies have reported similar positives [139,140]. However, a few studies also highlight contradicting findings which are multifactorial and scenario-dependent [141,142]. Studies assessing the use of telehealth diagnostics have also shown promising results in India which are similar to other studies conducted globally [126,127,143,144]. Our review also highlights the utility of digital health interventions in the overall education and skill training of HCPs. As shown by multiple studies conducted during the COVID-19 pandemic, remote learning facilitated by tele-education has proven to be an effective tool that can be harnessed even after the pandemic to make education and training more convenient and accessible [143,145–147].

## Implications of the findings

The need for decentralized healthcare planning was identified following the COVID-19 pandemic [7]. Our review identifies that with respect to telehealth, the generation of scientific literature on facilitators and barriers has been concentrated in a few states only. As the government is pushing for the digitization of healthcare through the Ayushman Bharat Digital Health Mission [148], it is important to understand the barriers and facilitators not just at the national level but also at the community level. More comparable evidence needs to be generated to understand the local factors affecting the implementation of telehealth in India. The findings from our review could guide future policy and help in generating policies that would not just be easier to implement but also be more effective. There is a need to take advantage of technology in healthcare and telehealth is a potential tool that can address poor access to healthcare in India. Upcoming policies should incorporate facilitators at the level of human resources, technical infrastructure, and software and address the barriers in these domains to take India closer to UHC with the help of telehealth. For example, formative research, use of local language, provision of training before introducing the intervention, and technical support to face difficulties while using the intervention were identified to be some of the facilitators that can be easily incorporated while planning health programs with digital health as a component. There is a need to improve the network coverage in areas implementing digital interventions. While designing the applications for the collection of data by community health workers, the potential overlap of information being collected should be avoided if multiple applications are being implemented through the same community health workforce to reduce workload. However, digital health interventions will only support the workforce and not replace them. Therefore there is a need to adequately recruit and train the HRH for proper service delivery from the healthcare system.

## Strengths and limitations

A few of the strengths of our studies are the use of a robust search strategy and the inclusion of a large number of studies. While previous reviews have assessed the overall utility of telemedicine, our review specifically looks at telehealth, which covers broader interventions, and its utility in the context of HCP providers. In the Indian context, the use of mHealth, telemedicine, and tele-education by community health workers has been an important highlight of our review. However, the findings of this review must be interpreted in the context of the following limitations. Firstly, facilitators and barriers could not be differentially studied between private

and public providers as the distinction between the type of provider could not be made using extracted data. Secondly, in this review we only provide a brief overview of the facilitators and barriers, and an in-depth analysis of study outcomes, meta-analysis, and critical appraisal of the risk of bias was not performed for the studies included. We listed sociocultural factors like poor literacy, stigma related to the disease, stigma related to technological intervention, and lack of human touch in the use of technology as barriers. However, a deeper understanding of how they impacted telehealth uptake was out of the scope of this paper and could be looked at in the upcoming reviews. Thirdly, while analyzing the number of studies from each state, data was not available for 28 studies, and 8 studies were conducted in multiple states with no mention of the names of the states involved. Fourthly, as this review is part of a larger evidence synthesis project on human resources for health, their management, and UHC, the primary aim of the search strategy was not to focus on telehealth. Therefore, telehealth-specific terms were not a part of the search strategy. Due to a broad search strategy, it is also possible that other less commonly used forms of telehealth like desktop and web-based telehealth interventions used by HCPs were missed in this review. However, despite not including these terms we were still able to extract information from a large volume of articles. Our review persists to be the single largest review on telehealth in India. Finally, due to a lack of funding for the project, we could not access databases like EMBASE and Scopus. This might have led to the exclusion of a few studies available on these databases. As mentioned previously, this review was a part of a larger set of reviews synthesized to inform the ongoing work of the Lancet Commission for Reimagining India's Health System. The choice for Pubmed was based on the fact that it is an open-access, and curated database of articles on health and medicine. We wanted to consider literature that passed the peer review check as the minimum requirement for scientific credibility and integrity to use them to inform the Commission confidently. Hence, sources of gray literature, such as Google Scholar, were avoided. Also, we wanted to ensure that the search is replicable for free around the world (open access). Hence, cost-prohibitive sources such as Web of Science and Scopus were avoided. However, the large volume of papers included adds to the comprehensiveness of our findings. Also, our strategy is replicable in other LMICs and resource-limited settings. We also included papers only published in the English language. However, as only the facilitators and barriers in India were being studied, English being the commonest language, the risk of language bias is unlikely in our opinion.

## Conclusion

The use of telehealth has not been studied uniformly across India. Systematic efforts need to be taken to anticipate and address barriers and implement telehealth intervention in ways to facilitate its uptake. Future studies should focus on looking at region-specific, intervention-specific, health-system (public vs. private), and health cadre-specific barriers and facilitators for the use of telehealth to promote decentralized decision-making for successfully implementing telehealth interventions in India.

## Supporting information

**S1 Checklist. PRISMA checklist.**
(DOCX)

**S1 Data. Data file.**
(XLSX)

**S1 Panel. PubMed search strategy.**
(DOCX)

**S2 Panel. Human resource cadres.**
(DOCX)

**S1 Table. Who action framework for human resources for health.**
(DOCX)

**S2 Table. Risk of bias assessment of included studies.**
(DOCX)

**S3 Table. Study characteristics for studies on mHealth as mentioned by the authors.**
(DOCX)

**S4 Table. Study characteristics for studies on telemedicine as mentioned by the authors.**
(DOCX)

**S5 Table. Study characteristics for studies on tele-education as mentioned by the authors.**
(DOCX)

## Acknowledgments

We want to thank Dr. Dipanwita Sengupta, Lancet Citizens' Commission Member for her support during the review process. We also want to thank Ms. Vashumathi Sriganesh from QMed Knowledge Foundation, Mumbai for providing her support in building the search strategy.

## Author Contributions

**Conceptualization:** Parth Sharma, Siddhesh Zadey, S. S. Prakash, Preethi John, Vikram Patel.

**Data curation:** Parth Sharma, Shirish Rao, Padmavathy Krishna Kumar, Aiswarya R. Nair, Gayathri Surendran, Rachna George Joseph, Girish Dayma, Liya Rafeekh, Shubhashis Saha, Sitanshi Sharma, S. S. Prakash, Venkatesan Sankarapandian.

**Formal analysis:** Parth Sharma, Disha Agrawal.

**Methodology:** Parth Sharma, Siddhesh Zadey, S. S. Prakash, Venkatesan Sankarapandian, Preethi John, Vikram Patel.

**Supervision:** Preethi John, Vikram Patel.

**Visualization:** Parth Sharma.

**Writing – original draft:** Parth Sharma, Shirish Rao, Padmavathy Krishna Kumar, Aiswarya R. Nair, Disha Agrawal.

**Writing – review & editing:** Siddhesh Zadey, Preethi John, Vikram Patel.

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
