## [Decision Letter · Decision Letter 0]

4 Dec 2023

PDIG-D-23-00391

Barriers and Facilitators for the Use of Telehealth by Healthcare Providers (HCP) in India - A Scoping Review

PLOS Digital Health

Dear Dr. Sharma,

Thank you for submitting your manuscript to PLOS Digital Health. After careful consideration, we feel that it has merit but does not fully meet PLOS Digital Health's publication criteria as it currently stands. Therefore, we invite you to submit a revised version of the manuscript that addresses the points raised during the review process.

Please submit your revised manuscript within 60 days Feb 02 2024 11:59PM. If you will need more time than this to complete your revisions, please reply to this message or contact the journal office at digitalhealth@plos.org. Please include the following items when submitting your revised manuscript:

We look forward to receiving your revised manuscript.

Kind regards,

Haleh Ayatollahi

Section Editor

PLOS Digital Health

Journal Requirements:

1. Please provide separate figure files in .tif or .eps format only and remove any figures embedded in your manuscript file. Please also ensure that all files are under our size limit of 10MB.

2. Some material included in your submission may be copyrighted. According to PLOS’s copyright policy, authors who use figures or other material (e.g., graphics, clipart, maps) from another author or copyright holder must demonstrate or obtain permission to publish this material under the Creative Commons Attribution 4.0 International (CC BY 4.0) License used by PLOS journals. Please closely review the details of PLOS’s copyright requirements here: PLOS Licenses and Copyright. If you need to request permissions from a copyright holder, you may use PLOS's Copyright Content Permission form.

Potential Copyright Issues:

Figs 2, 5 7 8: please (a) provide a direct link to the base layer of the map (i.e., the country or region border shape) and ensure this is also included in the figure legend; and (b) provide a link to the terms of use / license information for the base layer image or shapefile. We cannot publish proprietary or copyrighted maps (e.g. Google Maps, Mapquest) and the terms of use for your map base layer must be compatible with our CC-BY 4.0 license. 

"

Additional Editor Comments (if provided):

Reviewers' comments:

Reviewer's Responses to Questions

**Comments to the Author**

1. Does this manuscript meet PLOS Digital Health’s publication criteria? Is the manuscript technically sound, and do the data support the conclusions? The manuscript must describe methodologically and ethically rigorous research with conclusions that are appropriately drawn based on the data presented.

Reviewer #1: Yes

Reviewer #2: Partly

2. Has the statistical analysis been performed appropriately and rigorously?

Reviewer #1: N/A

Reviewer #2: I don't know

3. Have the authors made all data underlying the findings in their manuscript fully available (please refer to the Data Availability Statement at the start of the manuscript PDF file)?

Reviewer #1: Yes

Reviewer #2: Yes

4. Is the manuscript presented in an intelligible fashion and written in standard English?

Reviewer #1: No

Reviewer #2: No

5. Review Comments to the Author

Reviewer #1: This study is a comprehensive scoping review of barriers and facilitators to telehealth use by healthcare practitioners in India. This study provides interesting findings with the potential to inform healthcare delivery in India.

General Comments: 

The use of acronyms is inconsistent and should be double-checked. Some acronyms are never defined, some are used only once, and some are defined multiple times. Additionally, make sure citations are used appropriately, there were a few sentences that were missing citations. The discussion section was mostly a repetition of the findings and would be strengthened by contextualizing the findings within the existing literature. The authors should also consider reviewing the discussion section to improve clarity and grammar.

Specific Comments:

On Page 4, when the authors discuss the costs of healthcare, is that to individuals or system costs?

On Page 5, access to health services in what way, affordability? Accessibility?

Page 5 line 105 to 109 could be reworded for clarity. 

Page 6, do the authors have a justification for the timeframe of the search?

Page 7, definitions are provided for mHealth and telemedicine in the background, but not for tele-education, how is this a distinct category for the authors. 

Page 7, can more description be provided on the analysis technique. 

Page 19 line 406, are the authors referring to healthcare delivery?

Page 20 line 427 -429, “An understanding of the facilitators and barriers emphasizes the need for understanding the same at multiple dimensions especially focusing on the facilitators and barriers related to Human resources for health (HRH), Infrastructure, and Technology” What are the authors referring to when the describe “understanding the same”

Page 20 line 432, do the authors mean “roadblock”?

Page 20 line 435-436. This sentence is clumsy. The conclusion they are drawing here is that because women have "lesser access" to mobile phones and technology, increasing network coverage must also consider gender equality. I do not see how these two things are related as the authors have described it. Gender equity is a consideration in technology use and access. How do the authors suggest gender equity be considered in network coverage increases? How are the authors defining network coverage? Does that include access to devices, or broadband Internet and cell coverage as it is typically used?

Page 21 451-453, “Initiatives like the ‘G-20 Digital Innovation Alliance’ show promise in encouraging digital health startups by providing grants, sponsorship, and collaboration opportunities in order to strengthen the telehealth scenario”. This feels out of place, did the review identify start ups as a facilitator. I am not disagreeing with this point, just need a little more justification for how it fits within the discussion of barriers and facilitators.

Reviewer #2: The manuscript's language should be clear, correct, and unambiguous. To enhance readability, eliminate unnecessary content while ensuring that the paper maintains clarity. Major modifications are required in terms of grammar, punctuation, coordinating conjunctions, articles, and spaces between words. E.g.:

• The manuscript was checked for plagiarism, and it found a high use of generative AI without any declaration of such use. Also, there is a huge difference in the writing pattern in different sections or paragraphs. Additionally, some areas are not properly cited.

• It’s noted that the used map figures are all subjected to copyright, and can not be used without a written permission.

• Please clarify the abbreviation (HCPs) in first mention and not in the title.

• Please clarify the abbreviation (HRH) just in first mention and not every time you mention. 

• Please clarify the abbreviation (mHealth) in first mention.

• In page 5, line 109, what do you mean by ‘HR’?

• … And there is many more, make sure to do proofreading.

The manuscript claims to be a comprehensive analysis of telehealth in the Indian health system, yet a critical examination reveals several methodological and conceptual shortcomings that cast doubt on the validity and generalizability of the findings. While the effort to explore the facilitators and barriers of telehealth is commendable, the study lacks the depth and rigor necessary for drawing robust conclusions.

• Please clarify the omission of telepharmacy in this study, while it is an important point for consideration, as a significant component of telehealth and contributing to overall healthcare efficiency. Its exclusion from the discussions in this manuscript leaves a notable gap in the comprehensiveness of the review. Please redo the study work and add telepharmacy.

• The manuscript tends to overly emphasize the positive impacts of telehealth without providing a balanced view. A more critical approach would involve exploring potential drawbacks, ethical considerations, and challenges faced by both healthcare providers and patients in adopting telehealth technologies.

• The review does not adequately explore cultural factors that might influence the adoption and acceptance of telehealth in India.

• State which states were included in the review and if there were any reasons for the concentration in those particular regions. The study acknowledges a concentration of telehealth interventions in only 8 states/union territories, highlighting a significant regional bias.

• Discuss any potential differences or challenges related to the utilization of telehealth in the private sector, especially considering its accessibility to different socioeconomic strata.

• Mere mention of the G-20 Digital Innovation Alliance is not sufficient. A more critical discussion on the inadequacy of the current funding levels (2.1% of GDP) and potential consequences is warranted.

• Is there any comparative analysis or synthesis of findings across different studies, as identifying common themes or emerging patterns?

• The review provides a surface-level overview without a critical analysis of the study outcomes. A deeper examination of the methodologies, results, and implications of the included studies is essential for a more meaningful contribution.

• Relying solely on PubMed for the search strategy raises concerns about the comprehensiveness of the literature review. Exclusion of other databases like Scopus and EMBASE might result in a biased representation of the available evidence.

• Certain claims, such as the strong impact of telehealth on patient care and the assertion that it reduced travel constraints, lack concrete evidence or references. These statements should be supported by robust data to enhance the credibility of the review.

6. PLOS authors have the option to publish the peer review history of their article (what does this mean?). If published, this will include your full peer review and any attached files.

**Do you want your identity to be public for this peer review?** For information about this choice, including consent withdrawal, please see our Privacy Policy.

Reviewer #1: No

Reviewer #2: No

---

## [Decision Letter · Decision Letter 1]

22 Feb 2024

PDIG-D-23-00391R1

Barriers and Facilitators for the Use of Telehealth by Healthcare Providers in India - A Scoping Review

PLOS Digital Health

Dear Dr. Sharma,

Thank you for submitting your manuscript to PLOS Digital Health. After careful consideration, we feel that it has merit but does not fully meet PLOS Digital Health's publication criteria as it currently stands. Therefore, we invite you to submit a revised version of the manuscript that addresses the points raised during the review process.

Please submit your revised manuscript within 60 days Apr 22 2024 11:59PM. If you will need more time than this to complete your revisions, please reply to this message or contact the journal office at digitalhealth@plos.org. Please include the following items when submitting your revised manuscript:

We look forward to receiving your revised manuscript.

Kind regards,

Haleh Ayatollahi

Section Editor

PLOS Digital Health

Journal Requirements:

Additional Editor Comments (if provided):

Reviewers' comments:

Reviewer's Responses to Questions

**Comments to the Author**

1. If the authors have adequately addressed your comments raised in a previous round of review and you feel that this manuscript is now acceptable for publication, you may indicate that here to bypass the “Comments to the Author” section, enter your conflict of interest statement in the “Confidential to Editor” section, and submit your "Accept" recommendation.

Reviewer #1: All comments have been addressed

Reviewer #3: (No Response)

Reviewer #4: All comments have been addressed

2. Does this manuscript meet PLOS Digital Health’s publication criteria? Is the manuscript technically sound, and do the data support the conclusions? The manuscript must describe methodologically and ethically rigorous research with conclusions that are appropriately drawn based on the data presented.

Reviewer #1: Yes

Reviewer #3: Yes

Reviewer #4: No

3. Has the statistical analysis been performed appropriately and rigorously?

Reviewer #1: N/A

Reviewer #3: N/A

Reviewer #4: No

4. Have the authors made all data underlying the findings in their manuscript fully available (please refer to the Data Availability Statement at the start of the manuscript PDF file)?

Reviewer #1: Yes

Reviewer #3: Yes

Reviewer #4: Yes

5. Is the manuscript presented in an intelligible fashion and written in standard English?

Reviewer #1: Yes

Reviewer #3: Yes

Reviewer #4: No

6. Review Comments to the Author

Reviewer #1: Thank you for addressing each of the reviewer comments thoughtfully. I have no other concerns.

Reviewer #3: 1. The digital library considered in the study is only PubMed. Large number of telehealth solutions are proposed on other database. Instead of relying on a single database, it would have been better to shirink the scope and assess studies from everywhere in formulated scope. 

2. I think telemedicine and tele-education should be part of the keyword list

3. Placement of references - putting the references at the middle or end of statements increases readability of the paper. For instance, mHealth, on the other hand, refers to applications or programs used on smartphones or tablets (1) preferebale than (1) mHealth, on the other hand, refers to applications or programs used on smartphones or tablets 

4. Acronyms need to be defined at their first in-text appearance. Take HRH, it is defined in its second appearance at page 6, though the acronym appeared at page 5. There are also undefined acronyms...e.g. LMICs 

5. All Inclusion and exclusion criteria should be explicitly stated instead of mentioning some of them and indicating presence of more criteria with word like "like", "included", etc

6. Where is search string/query? 

7. There are many non-mHealth web-based and desktop software as well as websites ehealth solutions/interventions that are reported in literature. Same is true for hardware solutions. The remove didn't address reports on such solution. Hence, this needs to be mentioned either in the introduction or limitation of the study section. 

8. Some reviews results could be better presented using tables/figures than as a paragraph. Example, roles of interventions. 

9. I expected more technical future works than proposing other scoping review. For instance, what should be done to exploit the advantages of telehealth

Reviewer #4: Many thanks for the chance to review this manuscript. I think the findings of this review have the potential of improving the healthcare system in a populated country like India, and possibly advance digital health adoption and research. However, I think the manuscript currently have some major flaws, particularly on the methods. The methods need to be more thorough and explicit that would allow for a replication. The use of a single database for a scoping review is a HUGE flaw that cannot possibly be compensated for the lack of resources. If this is a scoping review, then the authors should consider including other materials (unless they did not find any) beyond published peer-reviewed publications. My specific comments are indictated below. 

General comments

1. The second paragraph in the introduction is not logically connected with the third one. The authors should consider making a logical connection or better still, remove it as it’s a bit off track. 

2. 

Methods

1. The methods is not robust enough: the authors should provide more information on the search strategy. What search terms were used and for which databases (authors only mentioned PubMed). If the search strategy was based on a framework, the authors should provide a background to the WHO Action framework , the authors provide a background to that framework and explain how the framework was used in this review. As it is now, it is unclear how the authors conducted their search. 

2. The use of only one database is hugely problematic and its not something that can simply be acknowledged as a limitation. By using only PubMed, my believe is that a significant number of papers that meet the inclusion criteria but were not indexed in PubMed might have been excluded. 

3. The authors removed some 203 duplicates, even though they claimed to have used only PubMed and I am wondering how that happened. Is it theoretically impossible for a paper to be indexed twice in the same database. You only remove duplicates when you search multiple sources and then remove papers that might have been indexed across two or more databases. So if a single database was used, how come duplicates were removed ?

4. I am not too sure if this is a scoping review or a systematic review. If this is a scoping review, then the authors should include other materials (published or unpublished) beyond peer-reviewed research papers. As it stands now, the authors excluded materials that ordinarily would qualify for inclusion in a scoping review. 

5. The authors stated in the methods overview that the study adopted Arksey and O’Malley methodological framework. However, the methods outlined is a bit inconsistent with Arksey and O’Malley framework. 

Results

6. The screening processes for arriving at the final papers as shown on the Prisma Diagram appears problematic. At the screening stage, the authors removed 439 papers due to inappropriate study design. I am wondering how they determined the study design when they were screening titles and abstracts because you can only determine the appropriateness of a study design until you review the full text. The authors also removed papers at the screening stage based on other criteria that can only be determined in full text review. 

7. The results is a bit scanty: More information is needed to shed light on the barriers and facilitators

7. PLOS authors have the option to publish the peer review history of their article (what does this mean?). If published, this will include your full peer review and any attached files.

**Do you want your identity to be public for this peer review?** For information about this choice, including consent withdrawal, please see our Privacy Policy. 

Reviewer #1: No

Reviewer #3: None

Reviewer #4: No

---

## [Decision Letter · Decision Letter 2]

15 May 2024

PDIG-D-23-00391R2

Barriers and Facilitators for the Use of Telehealth by Healthcare Providers in India - A Systematic Review

PLOS Digital Health

Dear Dr. Sharma,

Thank you for submitting your manuscript to PLOS Digital Health. After careful consideration, we feel that it has merit but does not fully meet PLOS Digital Health's publication criteria as it currently stands. Therefore, we invite you to submit a revised version of the manuscript that addresses the points raised during the review process.

Please submit your revised manuscript within 60 days Jul 14 2024 11:59PM. If you will need more time than this to complete your revisions, please reply to this message or contact the journal office at digitalhealth@plos.org. Please include the following items when submitting your revised manuscript:

We look forward to receiving your revised manuscript.

Kind regards,

Haleh Ayatollahi

Section Editor

PLOS Digital Health

Journal Requirements:

Additional Editor Comments (if provided):

Reviewers' comments:

Reviewer's Responses to Questions

**Comments to the Author**

1. If the authors have adequately addressed your comments raised in a previous round of review and you feel that this manuscript is now acceptable for publication, you may indicate that here to bypass the “Comments to the Author” section, enter your conflict of interest statement in the “Confidential to Editor” section, and submit your "Accept" recommendation.

Reviewer #1: (No Response)

Reviewer #3: All comments have been addressed

Reviewer #4: (No Response)

Reviewer #5: All comments have been addressed

2. Does this manuscript meet PLOS Digital Health’s publication criteria? Is the manuscript technically sound, and do the data support the conclusions? The manuscript must describe methodologically and ethically rigorous research with conclusions that are appropriately drawn based on the data presented.

Reviewer #1: (No Response)

Reviewer #3: Yes

Reviewer #4: Partly

Reviewer #5: Yes

3. Has the statistical analysis been performed appropriately and rigorously?

Reviewer #1: (No Response)

Reviewer #3: Yes

Reviewer #4: N/A

Reviewer #5: N/A

4. Have the authors made all data underlying the findings in their manuscript fully available (please refer to the Data Availability Statement at the start of the manuscript PDF file)?

Reviewer #1: (No Response)

Reviewer #3: (No Response)

Reviewer #4: Yes

Reviewer #5: Yes

5. Is the manuscript presented in an intelligible fashion and written in standard English?

Reviewer #1: (No Response)

Reviewer #3: Yes

Reviewer #4: Yes

Reviewer #5: Yes

6. Review Comments to the Author

Reviewer #1: This study is a comprehensive systematic review of barriers and facilitators to telehealth use by healthcare practitioners in India. This study provides interesting findings with the potential to inform healthcare delivery in India.

General Comments: 

This study was originally presented as a scoping review but is now being positioned as a systematic review. This should be acknowledged as a limitation and deviation from the original protocol (with justification). However, just suggesting this review is now a systematic review because it did not follow scoping review guidelines is problematic. For example, while scoping reviews do not assess risk of bias, systematic reviews are generally required to assess and report risk of bias. While the risk of bias was “N/A” when it was a scoping review, you will need to justify why not risk a bias was done and acknowledge this as a limitation. 

“At Level 1, the articles were screened based on the title and abstract. The articles included by any one reviewer at Level 1 screening were moved to Level 2.” What guided the authors screening approach?

“The full text of all the articles in Level 2 was reviewed independently by two reviewers. After the full-text screening, articles were finally excluded or included only if both reviewers were in agreement.” Can the authors provide inter rater reliability score for level 2 screening?

Reviewer #3: My comments are adequately addressed

Reviewer #4: Thank you for the opportunity to re-review this manuscript. While the authors provided their reasons for the choice of a single database, I don’t think those justification are founded, particularly for an interdisciplinary field like digital health, and here are my reasons. 

• The authors indicated that they choose only Pubmed because “it is open access”. PubMed is not necessarily an open-access databases because it only provides free access to the title and abstract and not full text. Full texts are available in PubMed ONLY if they are published as such in their respective journals. If a paper is not published as open-access, PubMed does not index that paper as a full-text in its database. If cost is indicated as a limitation, I am wondering how the authors retrieved the full text for articles that have free access to only titles and abstract in PubMed? I just searched for some of the included articles and beyond the title and abstracts, PubMed does not provide access to the full text? So how did the authors access the full text if cost was indicated as a limitation?

• They also stated that one of the reasons for choosing only PubMed is because it is “a specialized and curated database”. I think this is the more reason why more databases are needed. Digital health is an interdisciplinary field and using a “specialized” database may not be able to capture the diverse literature in the field

• Assuming without admitting that only PubMed was used because it is open-access, I think I think there are co-authors with affiliations to Columbia, Duke university, Harvard, UCL etc that might have had access to all databases to help expand the search. Therefore, cost cannot be an adequate justification when other avenues weren’t explored 

• The authors also decided to revert to a systematic review based on my previous comment. While this is laudable, I think they should go beyond just changing the name from scoping review to a systematic review by appraising the quality of the evidence. Without a quality appraisal of the literature, it still does not qualify as a systematic review, in my view.

Reviewer #5: The authors have thoughtfully considered all comments and have addressed them where possible. While there are some limitations in the review's methodology, primarily the use of only one database for searching, the authors have explained in detail why this is the case. The search strategy was also previously registered. This review provides useful look at how Indian healthcare is using mHealth applications.

7. PLOS authors have the option to publish the peer review history of their article (what does this mean?). If published, this will include your full peer review and any attached files.

**Do you want your identity to be public for this peer review?** For information about this choice, including consent withdrawal, please see our Privacy Policy. 

Reviewer #1: No

Reviewer #3: None

Reviewer #4: No

Reviewer #5: No

---

## [Decision Letter · Decision Letter 3]

2 Sep 2024

PDIG-D-23-00391R3

Barriers and Facilitators for the Use of Telehealth by Healthcare Providers in India - A Systematic Review

PLOS Digital Health

Dear Dr. Sharma,

Thank you for submitting your manuscript to PLOS Digital Health. After careful consideration, we feel that it has merit but does not fully meet PLOS Digital Health's publication criteria as it currently stands. Therefore, we invite you to submit a revised version of the manuscript that addresses the points raised during the review process.

Please submit your revised manuscript within 60 days Nov 01 2024 11:59PM. If you will need more time than this to complete your revisions, please reply to this message or contact the journal office at digitalhealth@plos.org. Please include the following items when submitting your revised manuscript:

We look forward to receiving your revised manuscript.

Kind regards,

Haleh Ayatollahi

Section Editor

PLOS Digital Health

Journal Requirements:

Additional Editor Comments (if provided):

Reviewers' comments:

Reviewer's Responses to Questions

**Comments to the Author**

1. If the authors have adequately addressed your comments raised in a previous round of review and you feel that this manuscript is now acceptable for publication, you may indicate that here to bypass the “Comments to the Author” section, enter your conflict of interest statement in the “Confidential to Editor” section, and submit your "Accept" recommendation.

Reviewer #3: All comments have been addressed

Reviewer #4: (No Response)

Reviewer #6: All comments have been addressed

Reviewer #7: (No Response)

2. Does this manuscript meet PLOS Digital Health’s publication criteria? Is the manuscript technically sound, and do the data support the conclusions? The manuscript must describe methodologically and ethically rigorous research with conclusions that are appropriately drawn based on the data presented.

Reviewer #3: Yes

Reviewer #4: No

Reviewer #6: Partly

Reviewer #7: Yes

3. Has the statistical analysis been performed appropriately and rigorously?

Reviewer #3: Yes

Reviewer #4: N/A

Reviewer #6: (No Response)

Reviewer #7: Yes

4. Have the authors made all data underlying the findings in their manuscript fully available (please refer to the Data Availability Statement at the start of the manuscript PDF file)?

Reviewer #3: No

Reviewer #4: Yes

Reviewer #6: No

Reviewer #7: Yes

5. Is the manuscript presented in an intelligible fashion and written in standard English?

Reviewer #3: Yes

Reviewer #4: Yes

Reviewer #6: Yes

Reviewer #7: Yes

6. Review Comments to the Author

Reviewer #3: My comments are addressed

Reviewer #4: Many thanks for the opportunity to re-review this manuscript. While the authors might have addressed almost all comments, I think the substantive concern remains unaddressed and I don’t think the explanations helps either. The use of a single database is a huge flaw that cannot simply be discounted as a study limitation. Without a robust literature search, I see this as someone trying to put up a house on a week foundation. The issue of cost would have been totally acceptable if it probably led the authors to limit the geographical scope of the review or to tie it to specific types of digital health technologies. However, limiting the methodological scope, with all due respect, falls short of the standards required for systematic reviews. In systematic reviews, searching is one thing and retrieving the literature is another thing. You can conduct searchers in PubMed but cannot retrieve the literature from PubMed, unless they are open access. I don’t think all the 2944 articles were open access and it would be erroneous for institutions to subscribe to individual journals (where you were able to retrieve this tone of literature) and not subscribe to publishing companies where they can have access to many journals as possible. 

I want to author to take this one main criticism in good faith. I am not being critical for the sake of it but to make sure that the eventual publication will be robust enough and meet the rigor expected of a systematic review.

Reviewer #6: Dear Authors,

Thank you for the opportunity to review your manuscript on telehealth use by healthcare providers in India. I found your work to be a valuable contribution to the field, addressing a timely and important topic. Your systematic approach and comprehensive analysis of 106 studies provide a solid foundation for understanding the landscape of telehealth in India.

I particularly appreciated your thorough examination of facilitators and barriers across mHealth, telemedicine, and tele-education. This multi-faceted approach offers valuable insights for both practitioners and policymakers.

However, I believe there are several areas where your manuscript could be strengthened:

Firstly, I noticed the absence of a Data Availability Statement. As per PLOS policy, this is a crucial element that needs to be addressed.

Regarding your methodology, while your search strategy is comprehensive, it's limited to PubMed. I'd encourage you to consider expanding your search to include other databases like Embase or Web of Science, or provide a rationale for their exclusion in the or clear statement explaining, i know you have alrady explained this, but clarify in the narrtive.

The risk of bias assessment could benefit from more detailed explanation. A summary table of these assessments would be a valuable addition to your manuscript.

Your analysis, while thorough, could be deepened in certain areas. For instance, a more quantitative synthesis of findings, where possible, would enhance the impact of your work. Additionally, further exploration of differences between public and private providers, and more discussion on regional variations would provide richer insights.

I'd also suggest expanding your discussion of limitations to include potential language bias and the lack of grey literature inclusion. Your policy implications section could be more specific, offering concrete recommendations based on your findings.

Lastly, some of your figures (particularly Figures 2, 5, and 8) could be improved for clarity. Consider using color gradients or larger font sizes for state names.

Overall, your manuscript is a strong piece of work. Addressing these points will further enhance its quality and impact. I look forward to seeing the revised version.

Best regards,

The Reviewer

Reviewer #7: The manuscript seems to address all previous reviewer comments. It presents an interesting work pertinent to the Indian healthcare system. Coming from a background where I also work in telemedicine for LMICs, the findings of this review seem to align greatly with the challenges and opportunities we experience. I think the work is beneficial not only for Indian contexts, but in general for developing healthcare systems. I do not have any comments regarding further improvements at this time.

7. PLOS authors have the option to publish the peer review history of their article (what does this mean?). If published, this will include your full peer review and any attached files.

**Do you want your identity to be public for this peer review?** For information about this choice, including consent withdrawal, please see our Privacy Policy. 

Reviewer #3: None

Reviewer #4: No

Reviewer #6: No

Reviewer #7: No

---

## [Editor Report · Decision Letter 4]

9 Oct 2024

Barriers and Facilitators for the Use of Telehealth by Healthcare Providers in India - A Systematic Review

PDIG-D-23-00391R4

Dear Dr Sharma,

We are pleased to inform you that your manuscript 'Barriers and Facilitators for the Use of Telehealth by Healthcare Providers in India - A Systematic Review' has been provisionally accepted for publication in PLOS Digital Health.

Best regards,

Haleh Ayatollahi

Section Editor

PLOS Digital Health